



**Evaluation of cloud effects on air temperature estimation using MODIS LST**
**based on ground measurements over the Tibetan Plateau**
Hongbo Zhang[1, 2], Fan Zhang[1, 2], Guoqing Zhang[1, 2], Xiaobo He[3], Lide Tian[1, 2]
[1] Key Laboratory of Tibetan Environment Changes and Land Surface Processes, Institute of
Tibetan Plateau Research, Chinese Academy of Sciences, Beijing, China
[2] CAS Center for Excellence in Tibetan Plateau Earth Sciences, Beijing, China
[3] Cold and Arid Regions Environmental and Engineering Research Institute, Chinese Academy of
Sciences, Lanzhou, China
*Correspondence to*: Hongbo Zhang, Tel.: +86-10-84097030; fax: +86-10-84097079.
E-mail address: zhanghongbo@itpcas.ac.cn





**Abstract**
Moderate Resolution Imaging Spectroradiometer (MODIS) land surface temperature (LST) data
have played a significant role in estimating the air temperature ($T_{air}$) due to the sparseness of
ground measurements, especially for remote mountainous areas. Generally, two types of air
temperatures are studied including daily maximum ($T_{max}$) and minimum ($T_{min}$) air temperatures.
MODIS daytime and nighttime LST are often used as proxies for estimating $T_{max}$ and $T_{min}$,
respectively. The Tibetan Plateau (TP) has a high daily cloud cover fraction (>45%). The presence
of clouds can affect the relationship between $T_{air}$ and LST and can further affect the estimation
accuracies. This study comprehensively analyzes the effects of clouds on $T_{air}$ estimation based on
MODIS LST using detailed half-hourly ground measurements and daily meteorological station
observations collected from over the TP. Comparisons made between in-situ cloudiness
observations and MODIS claimed clear-sky records show that erroneous rates of MODIS
nighttime cloud detection are obviously higher than those achieved in daytime. Our validation of
the MODIS LST values under different cloudiness constraining conditions shows that the
accuracy of MODIS nighttime LST is severely affected by undetected clouds. Large errors
introduced by undetected clouds are found to significantly affect the $T_{min}$ estimations based on
nighttime LST through cloud effect tests. However, clouds are mainly found to affect $T_{max}$
estimation by affecting the essential relationship between $T_{max}$ and daytime LST. The obviously
larger errors of $T_{max}$ estimation than those of $T_{min}$ could be attributed to larger MODIS daytime
LST errors resulting from higher degrees of daytime LST heterogeneity within MODIS pixel than
those of nighttime LST. Constraining all four MODIS observations per day to non-cloudy
observations can efficiently screen samples to build a strong fit of $T_{min}$ estimation using MODIS
nighttime LST. The present study reveals the effects of clouds on $T_{air}$ estimation through MODIS
LST and will thus help improve the estimation accuracy levels while alleviating the problems
associated with severe data sparseness over the TP.

**Keywords:** cloud effects, MODIS LST, air temperature estimation, Tibetan Plateau



**1 Introduction**

Air temperature is a key variable used to describe environmental conditions. However, temperature observations are typically sparse in remote mountainous areas (Lin et al., 2016). Remotely sensed land surface temperatures (LST) can serve as an efficient proxy for air temperature estimation in such areas. Superior to limited ground measurements, remote sensing can provide more spatiotemporal information. Several studies have estimated air temperatures using Moderate Resolution Imaging Spectroradiometer (MODIS) land surface temperature products for Europe (Kilibarda et al., 2014;Benali et al., 2012), Canada (Xu et al., 2014), USA (Oyler et al., 2016;Oyler et al., 2015;Parmentier et al., 2015), Africa (Vancutsem et al., 2010;Lin et al., 2012), western Asia (Emamifar et al., 2013) and the Tibetan Plateau (TP) (Zhu et al., 2013;Fu et al., 2011).

Due to its high altitudes, the TP includes the largest cryosphere area outside the Arctic and Antarctic regions and outside Greenland, and it is considered to be among the areas that are most sensitive to climate change. However, most meteorological stations in the TP are located in low-altitude (< 4800 m) and eastern regions (Fig. 1). There are almost no stations in the vast western area or at the elevations above 5000 m. In particular, for glacier covered areas, temperature observations are extremely scarce (Wu et al., 2015). Remotely sensed LSTs can greatly help alleviate the problems associated with scarce temperature observations available for the TP.

Despite the advantages of high spatial and temporal accessibility to large-scale areas, remote sensing data present some limitations, among which cloud contamination issues may be the most important. For applications of MODIS LST, clouds can affect the $T_{air}$ estimation in at least two ways: erroneous cloud identification can reduce the accuracy of MODIS LST values, and the presence of clouds can affect the relationship between LST and $T_{air}$ and can further affect the accuracy of $T_{air}$ estimations.

The presence of clouds can greatly decrease the amount of data available in the satellite images. Moreover, the existing cloud detection algorithms cannot identify all the cloudy pixels, and a considerable percentage of undetected cloudy pixels exists in MODIS LST products (reported at roughly 15%) (Ackerman et al., 2008). It has been shown through some validation studies that extremely large differences (>10 K) between MODIS LST and ground measurements occasionally





occur, even for homogeneous surfaces. In these cases, the cloud top temperatures can be taken as
the LST values (Langer et al., 2010;Westermann et al., 2011). More recently, up to 40% of ground
measured cloudy samples have been labeled unidentified according to field observations, thus
producing rather large MODIS LST errors, as reported for Svalbard (Østby et al., 2014). Such
errors can disturb the true relationship between LST and air temperatures ($T_{air}$). MODIS daytime
LST has been found to be affected by unidentified cloudy pixels, causing such pixels to severely
degrade LST-$T_{air}$ relationships (Williamson et al., 2013). Because the daytime cloud algorithm is
expected to present more confidence than that for nighttime (Ackerman et al., 1998), using the
nighttime LST for air temperatures may be influenced more by undetected clouds. For the TP,
cloud contamination also constitutes a major problem, generating a mean daily cloud cover
fraction of > 45% (Yu et al.). Thus, the effects of clouds are particularly essential for $T_{air}$
estimation in the TP.
In addition to the effects of undetected cloudy pixels, clouds are expected to play a key role in the
relationship between LST and $T_{air}$ due to its cooling effects during the day and warming effects at
night (Dai et al., 1999). During the day, clouds can decrease land surface warming rates by
blocking solar radiation, and at night, clouds can reflect surface long wave radiation and decrease
heat losses from the land surface producing higher ground temperatures than those detected on
clear days. For example, the difference between observed daytime LST and $T_{air}$ under cloudy
conditions is much lower (an average of ~3.7 °C) than that observed under clear conditions (Gallo
et al., 2011). Therefore, questions regarding whether and how clouds can affect relationships of
$T_{max}$-Daytime LST and $T_{min}$-Nighttime LST have been posed. Previous $T_{air}$ estimation based on
MODIS LST are presumably valid for clear conditions (Shen and Leptoukh, 2011;Oyler et al.,
2015). However, satellite observed LSTs (in night or day) are instantaneous and may have a time
lag between the overpass time and the time when $T_{air}$ reaches its minimum or maximum. Daily
cloudiness conditions may affect the warming (during the day) or cooling (at night) rates and can
further alter the relationship between $T_{air}$ and LST.
Previous studies have mainly focused on two types of daily $T_{air}$ estimations: daily maximum ($T_{max}$)
and minimum ($T_{min}$) air temperatures (Xu et al., 2014;Benali et al., 2012;Good, 2015). In addition,
daytime and nighttime LST have been used as predictors for $T_{max}$ and $T_{min}$ estimations,
respectively, due to their different overpass times (Vancutsem et al., 2010;Lin et al., 2012;Oyler et



al., 2016). Recent studies have interestingly found that the estimation accuracy of $T_{max}$ based on
daytime LST is clearly lower than that of $T_{min}$ based on nighttime LST (Oyler et al., 2016;Benali
et al., 2012;Zhang et al., 2011), and nighttime LST has an even higher correlation with $T_{max}$ than
daytime LST (Zeng et al., 2015;Zhang et al., 2011). Benali et al. (2012) hypothesized that the
presence of cloud cover may decrease daytime warming levels, resulting in incorrect modeling
and negative effects of cloud cover on estimation accuracies. Oyler et al. (2016) instead attributed
this to the large microscale variability differences between daytime and nighttime LST.
Due to the scarcity of detailed cloud observations available, few studies have focused on the
potentially important effects of clouds on estimations of $T_{air}$ using remotely sensed LST. This
study explores the effects of clouds on $T_{air}$ estimation using MODIS LST based on detailed
half-hourly ground measurements and the daily China Meteorological Administration (CMA)
station observations. For the TP, sufficiently detailed observations are extremely rare and related
studies have not been conducted before. Three automatic weather stations (AWS) with half-hourly
averaged observations are examined in this study, including one valuable site positioned on the
glacier. To make our study more representative, data drawn from 92 CMA stations that include
daily $T_{max}$ and $T_{min}$ observations are also used for cloud effect tests.
**2 Data**
**2.1 Ground measurements**
To conduct this study, detailed observations drawn from three AWSs on the TP were obtained (Fig.
1). The Ngari station is located in the western area of the TP at an elevation of 4270 m. Desert
grassland constitutes the main form of land cover here. The Qinghai station is located in the
northeastern TP at an elevation of 3250 m and is dominated by alpine meadow. The Xiao
Dongkemadi station is located in the interior TP at an elevation of 5621 m on the Xiao
Dongkemadi glacier (Fig. 1). The general features of the three AWSs are listed in Table 1. In
addition, observations drawn from 92 CMA stations over the TP are used for our assistant
analysis.
All three AWSs provide half-hourly averaged ingoing and outgoing longwave radiation, and air
temperature data. These radiation data were measured using a widely used CNR1 net radiometer,
at an uncertainty level of ±10% for daily totals by the manufacture. Air temperatures were
collected using an HMP45C sensor with expected accuracies of ±0.2–0.5 °C depending on the





temperature ranges involved. Detailed measurement specifications are listed in Table 1. However,
only the Xiao Dongkemadi station provides the directly measured LST values which were
obtained through an Apogee Precision Infrared Thermocouple Sensor (IRTS-P) with an accuracy
of 0.3 K over the glacier surface (Huintjes et al., 2015). The LSTs of the Qinghai and Ngari
stations were derived based on the thermal radiative transfer theory:

$$L_u = \sigma T_b^4 = (1 - \varepsilon)L_d + \varepsilon \sigma T_s^4$$

where $L_u$ and $L_d$ are the upwelling and downwelling longwave radiation, respectively, $\sigma$ is the
Stefan–Boltzmann constant ($5.670367 \times 10^{-8}$ Wm$^{-2}$ K$^{-4}$), $\varepsilon$ is land surface emissivity, $T_b$ is the
brightness temperature, $T_s$ is the land surface temperature.
In this study, emissivity values were assigned empirically due to a lack of measurements.
Emissivity values for the Qinghai and Ngari stations were set to 0.987 (alpine meadow) and 0.975
(desert grassland), respectively, according to Wang et al. (2008). To partly quantify the effects of
emissivity value uncertainty, simple sensitivity tests were conducted. A 0.001 change in emissivity
is on average found to result in the LST change of 0.015 K and 0.020 K for stations Qinghai and
Ngari, respectively.
Through controlling the data quality did by the data provider, obvious outliers have been removed
for all three AWSs. In addition, the 92 CMA stations provide daily $T_{max}$ and $T_{min}$ observations
measured at 2 m above the ground surface. Data drawn from these CMA stations are for 2007 to

153    2010.


**2.2 MODIS Land Surface Temperatures**

Daily 1-km LST products of MODIS level 3 collection 5 are used in this study including the data
from the Terra (MOD11A1) and Aqua (MYD11A1) satellites. Both Terra and Aqua generate two
daily observations, including one for the daytime and one for nighttime. The two overpass times
for Aqua are approximately 1:30 and 13:30 local time. For Terra, these times are approximately
10:30 and 22:30. Accurate view times can be derived from the product. The MODIS LST used
here is retrieved using the generalized Split-window algorithm (Wan and Dozier, 1996).
Accuracies are reported to range within 1 K, but the uncertainties and errors of emissivity used in
the MODIS LST product can be significant, which produces major errors (Wan et al., 2002). Each
grid of the MODIS LST product includes a quality control (QC) flag that ranges from 0 to 3
indicating the average errors of <1 K, 1−2 K, 2−3 K and >3 K. Records with a QC flag of 3 were
omitted in this study.
The MODIS observations are instantaneous, whereas the ground measurements used are
half-hourly averaged. To make them comparable, the timing of ground observations recorded on
Beijing time was converted to local solar time. Then, half-hourly observations that are within 15
minutes of the view times of MODIS record times were selected.

**3 Methods**
**3.1 Cloud index estimations**
Cloud observations are usually only available from non-automatic weather stations and are
difficult to record. In this study, an efficient method was employed to estimate cloudiness based on
downwelling longwave radiation ($L_d$) records and air temperatures, which have been widely used
in other studies (Yang et al., 2011;Østby et al., 2014;Giesen et al., 2008). This theory is mainly
based on the principle that under cloudy conditions, a longwave radiation balance exists between
cloud base and near surface (Østby et al., 2014;Giesen et al., 2008). Under overcast conditions,
both the cloud base and near surface radiate at similar temperatures and $L_d$ reaches its max.
However, $L_d$ should be much lower under clear conditions than under overcast conditions under
the same temperature. In such a case, $L_d$ reaches its minimum. Thus, a max $L_d$ can be reversed
using the Stefan–Boltzmann law under a given air temperature, and the min $L_d$ can be regressed
using the polynomial fit of the lower 5th percentile of the $L_d$ observations for each specified
temperature interval (1 K here). When $L_d$ is assumed to linearly increase from clear to overcast
conditions at a given temperature, then a "cloud index" (CI) indicating the cloudiness can be
achieved (CI = 0 and 1 for clear and overcast skies respectively). Rather than the visually
observed percentage of cloud cover in the sky, the CI used here represents the optical thickness of
clouds (Van Den Broeke et al., 2006).

**3.2 Testing cloud effects on the accuracies of MODIS LST**
Undetected clouds may exist in the MODIS LST data as a result of erroneous cloud identification.
An evaluation of the number of undetected clouds present was firstly conducted. As considerable
errors can be introduced by undetected clouds, the effects of clouds on MODIS LST accuracies





were evaluated by comparing validation (MODIS vs. observed LST) results derived before and
after removing the undetected cloudy records. In this study, the records with CI > 0.5 are
considered to be under "mostly cloudy" conditions. For a given MODIS observation, it is regarded
as undetected cloud if its corresponding CI > 0.5.
In our study, all four MODIS observations drawn from the Terra and Aqua satellites were
validated to identify and explain the effects of clouds on $T_{air}$ estimations.

**3.3 $T_{air}$ estimation**
A simple linear regression was used for $T_{air}$ estimations. This method has been the most commonly
used in previous studies (Benali et al., 2012;Lin et al., 2012;Zhang et al., 2011). Although more
advanced models, including neural network (Jang et al., 2004), random forests (Xu et al., 2014)
and M5 model tree (Emamifar et al., 2013), can be more accurate, they can also introduce
uncertainties owing to their much larger number of parameters. Because an individual linear fit is
built for each AWS or CMA station, variables indicating spatial coordinates (longitudes and
latitudes) and land cover (e.g. NDVI) are not used. Thus, only LST is selected as the independent
variable for $T_{air}$ regression. This is also why the machine learning methods and geostatistical
models (Oyler et al., 2015;Kilibarda et al., 2014) which generally involve the use of multiple
variables, are not used in this study.

**3.4 Testing cloud effects by the observed LST**
Large MODIS LST errors may exist due to undetected clouds, and cloud effects are first tested
using the ground measured LST. In this way, we can explore the direct effects of clouds on $T_{air}$
estimation using LST. The tests are conducted by constraining cloudiness conditions. Target $T_{air}$
values in most studies are daily (max, mean or min) values, but instantaneous cloudiness is
meaningless. In this study, the daily mean CI value is used as a cloudiness indicator. To ensure a
sufficient number of samples, 9 types of conditions with daily mean CI values ≤ 0.2, 0.3, …, 0.9
and 1.0 are employed, indicating that the cloudiness constraints vary from highly clear conditions
(daily mean CI ≤ 0.2) to fully mixed conditions, with many highly cloudy days included (daily
mean CI ≤ 1.0). For each condition, $T_{max}$ and $T_{min}$ are regressed using daytime (13:30, Aqua) and
nighttime (22:30, Terra) observed LST through a simple linear regression, and estimation





accuracies are computed. The root-mean-square error (RMSE) and mean absolute error (MAE) are
used as the accuracy measurements. Cloud effects are evaluated based on the variation of the
estimation accuracies under different cloudiness conditions. Comparisons of $T_{max}$ and $T_{min}$
estimations can reveal further implications of cloud effects.

**3.5 Determining cloud effects through comparisons using MODIS and the observed LST**
Once the effects of clouds on $T_{air}$ estimations using observed LST are confirmed, cloud effects on
$T_{air}$ estimation using MODIS LST can be explored more directly. Apart from affecting the
relationship between $T_{air}$ and MODIS LST, clouds can degrade the MODIS LST accuracy and
further reduce estimation accuracies. Such effects, when they are present, can be explored by
comparing changes in estimation accuracy levels between observed LST and MODIS LST. Here,
$T_{air}$ ($T_{min}$ and $T_{max}$) estimations for 9 kinds of CI conditions are conducted using MODIS LST and
observed LST (at the corresponding MODIS time), respectively. The results are analyzed based on
comparisons.

**3.6 Exploring cloud effects based on observations from meteorological stations**
In practice, only daily observations can be easily obtained from meteorological stations, and
cloudiness observations are usually not provided. In this study, only daily $T_{max}$ and $T_{min}$ data are
obtained from the 92 CMA stations. Nonetheless, daily cloudiness levels can be partly evaluated
from four MODIS observations for each day (two from Terra and two from Aqua). Then,
comparisons of $T_{air}$ estimation for two distinct cloudiness conditions are drawn.
Two conditions ("cloudy day" and "non-cloudy day") are defined based on four instantaneous
MODIS observations for each day for both the $T_{max}$ and $T_{min}$ estimation using Aqua daytime LST
and Terra nighttime LST, respectively. For "non-cloudy day" conditions, all four MODIS
cloudiness observations are constrained as non-cloudy. For the "cloudy day" condition of the $T_{max}$
estimation, Aqua daytime observations are constrained as non-cloudy to obtain the available LST,
and Terra daytime observations are constrained as cloudy to make cloud effects as strong as
possible. However, the Aqua night and Terra night observations are not constrained to obtain
sufficient samples. For the "cloudy day" condition of the $T_{min}$ estimation, the Terra nighttime
observations are constrained as non-cloudy to obtain the available LST, whereas the Aqua



nighttime observations are not constrained to obtain sufficient samples. Both Aqua daytime and
Terra daytime observations are constrained as cloudy to make the cloud effects as strong as
possible. $T_{max}$ and $T_{min}$ estimation accuracies are then compared under "cloudy day" and
"non-cloudy day" conditions.

**4 Result**
**4.1 Cloud index estimation and the undetected clouds of MODIS**
Figure 2 shows that the maximum and minimum $L_d$ curves effectively frame $L_d$ variation for each
air temperature. The CI values of all of the observations are then computed.
For each of the four overpass times of MODIS LST, a rate of undetected cloudy records can be
determined using CI values (Table 2). The ratio of undetected cloudy records ranges from 3% to
50% with a fully averaged ratio of 15%. This agrees well with the reported value of ~15%, which
was computed based on a consistency comparison between MODIS and Lidar (Ackerman et al.,

268   2008).


**4.2 MODIS LST validation under different cloud conditions**
The accuracy of MDOIS LST can be affected by undetected cloudy pixels (Shamir and
Georgakakos, 2014;Westermann et al., 2012). Figure 3 shows that after removing cloudy cases,
the validation accuracies of all three sites present obviously lower MAE values and a better fit line
slope. Improvements in accuracy for 6 (2 pass times × 3 stations) nighttime cases range from 0.1
to 0.9 °C. However, no significant accuracy improvements were found after removing cloudy
cases for daytime MODIS LST (Fig. 4). Only slightly better or comparative MAEs ($\leqslant$0.1 °C )
were obtained.
This indicates that the accuracy of MODIS nighttime LST is more negatively affected by
undetected clouds than that for the daytime. The relatively weak influences of undetected clouds
on daytime LST is mainly due to obviously lower erroneous rates of cloud detection compared to
those of nighttime LST. Erroneous rates of MODIS nighttime cloud detection are clearly larger
than those for the daytime, though not in the case of the Terra LST observed for Ngari. This can be
largely attributed to differences in cloud detection methods used for the daytime and nighttime.
The cloud detection algorithm of MODIS is considered to present more confidence for the



daytime than for the nighttime due to the absence of reflected solar radiation during nighttime
(Ackerman et al., 1998). This finding is consistent with previous studies showing that more than
40% of the observed cloudy days are identified as clear days by MODIS at polar summer
nighttime (Østby et al., 2014).

**4.3 The effects of clouds on $T_{air}$ estimation based on ground observed LST**
Figure 5 shows the accuracy of $T_{air}$ estimations based on ground observed LST under different
cloudiness conditions across the three sites. For $T_{max}$, estimation errors including RMSE and MAE
continually increased as the cloudiness condition constraints eased. The increase in RMSE/MAE
values for clear conditions (daily mean CI $\leqslant$ 0.2) compared with totally mixed conditions (daily
mean CI $\leqslant$1) was 1.3 °C/1.0 °C, 0.8 °C/0.8 °C and 1.6 °C/1.6 °C for the Ngari, Xiao
Dongkemadi and Qinghai stations, respectively. In contrast, for $T_{min}$, accuracy variation is
consistently mild across the three sites, presenting RMSE/MAE changes of 0.1 °C/0.0 °C,
0.1 °C/0.0 °C, and 0.7 °C/0.6 °C for the Ngari, Xiao Dongkemadi and Qinghai stations,
respectively.
As expected for cases based on ground observed LST, the $T_{max}$ estimation is significantly affected
by cloud conditions, but clouds have a limited effect on the $T_{min}$ estimation compared to $T_{max}$. This
interesting finding can be explained by mechanisms through which clouds affect nighttime and
daytime surface temperatures. In the daytime, LST is significantly influenced by solar heating.
The presence of clouds can screen out solar radiation and cool the surface. Much larger
differences between LST and $T_{air}$ have been observed under cloudy days than under clear
conditions (Gallo et al., 2011). At night, the surface can also present warming effects from clouds
due to reflected infrared longwave radiation. However, such effects are not typically significant
because the net effect of clouds on surface downward longwave radiation is much less pronounced
than nighttime solar cooling effects in most cases, as indicated by Dai et al. (1999).

**4.4 The effects of clouds on $T_{air}$ estimation based on MODIS LST**
Figure 6 compares cloud effects on $T_{min}$ and $T_{max}$ estimations using MODIS and observed LST.
First, despite rather mild effects of cloud conditions on $T_{min}$ estimation based on ground observed
LST, those based on MODIS LST are clearly much more significant. For cases based on MODIS





LST, increases in RMSE between clear (daily mean CI $\leqslant$ 0.2) and mixed conditions (daily mean
CI $\leqslant$ 1.0) are 0.5, 0.8, and 1.8 °C for the Ngari, Xiao Dongkemadi and Qinghai stations,
respectively. However, those for cases based on observed LST are significantly lower with
corresponding values of 0.0, -0.1, and 0.2 °C.
This indicates that $T_{min}$ estimations based on MODIS LST are greatly affected by clouds. This
seems counterintuitive, as it has been shown that $T_{min}$ estimations based on ground observed LST
are not significantly affected by clouds (Fig. 5). Thus, the most probable driving factor may be the
relatively large amounts of undetected clouds present in MODIS nighttime LST. As daily cloud
indexes increase, more undetected cloudy cases may be introduced, thus reducing the accuracy of
MODIS nighttime LST (Fig. 3 and Table 2).
Figure 7 (upper section) supports this conclusion: under clear conditions, the undetected clouds
are rare, and limited accuracy improvements are achieved by removing the few cloudy MODIS
LST records; However, as daily CI constraints ease to 0.5 when cloudy records account for a
substantial proportion, obvious improvements appear, and the final accuracies are much closer to
and are even better than those based on ground observed LST.
Unlike that of $T_{min}$, the accuracy variation of $T_{max}$ estimation based on MODIS LST shows trends
that are highly consistent with those of cases based on ground observed LST for all of the three
sites. As with cases based on ground observed LST, $T_{max}$ estimation based on MODIS LST are
found to be greatly affected by clouds. In addition, increases in ($T_{max}$ estimation based on MODIS
LST vs. that based on ground observed LST) in accuracy level differences between clear and
mixed conditions are much less pronounced compared to those of $T_{min}$, where difference values
are only 0.0, 0.2 and 0.3 °C for the Ngari, Xiao Dongkemadi and Qinghai stations, respectively.
However, the accuracy levels achieved from MODS LST after removing cloudy records are
obviously lower than those based on ground observed LST under all cloudiness conditions. This
raises questions regarding what this difference in accuracy attribute to? Dominant factors may not
be undetected clouds, as was the case for $T_{min}$. As shown in Fig. 7 (lower section), the removal of
cloudy records had somewhat moderate effects on accuracy levels. This may be largely due to
much lower erroneous rates of cloud identification for MODIS daytime LST. The obviously lower
number of undetected clouds compared to nighttime LST values for the Ngari and Qinghai
stations result in relatively limited accuracy improvements. The relatively large decrease in



estimation errors for the Xiao Dongkemadi station is mainly due to unexpected higher amounts of
undetected clouds in MODIS daytime LST for that site (Table 2 and Fig. 7).
Furthermore, even under clear conditions, the accuracy of $T_{max}$ estimations based on MODIS LST
is remarkably lower than that based on ground observed LST (Fig. 6). Thus, the decrease in
accuracy levels relative to cases based on ground observed LST may be caused by other factors
rather than undetected clouds. This seems odd, especially given that the accuracies of $T_{min}$
estimations based on MODIS LST are very close to or even better than those based on observed
LST under clear conditions (Fig. 6).

**4.5 Effects of clouds on $T_{air}$ estimation based on MODIS LST and CMA observations**
Figure 8 shows the estimation accuracies of $T_{air}$ based on MODIS LST for non-cloudy and cloudy
conditions. For the $T_{max}$ estimation, clouds appear to have moderate effects on estimation
accuracies, where 88% of the 92 stations obtained lower RMSEs based on samples from
"non-cloudy" conditions relative to cloudy cases. RMSE values are reduced by an average of
0.54 °C. In contrast, effects of clouds on $T_{min}$ estimations are much more significant: the RMSEs
of 98% stations are reduced by an average of 1.44 °C. Though hourly observations in the data for
CMA stations are lacking, the results for the cloud tests are highly consistent with those based on
half-hourly AWS observations.
Furthermore, a comparison between the $T_{max}$ and $T_{min}$ estimation results based on MODIS LST
and CMA observations shows that under cloudy conditions, $T_{max}$ estimations (the mean RMSE is
4.3 °C) achieve generally higher levels of accuracy than $T_{min}$ estimations (the mean RMSE is
4.6 °C), whereas non-cloudy conditions produce the opposite effect (3.7 vs. 3.2 °C) illustrating
potentially stronger negative effect of cloud on $T_{max}$ estimation than $T_{min}$ .

**5 Discussion**
**5.1 Differences in the effects of clouds on $T_{min}$ and $T_{max}$ estimations based on MODIS LST**
From MODIS LST and daily CMA observations, different cloud effects between $T_{max}$ and $T_{min}$
estimations can be identified from Fig. 8. Under cloudy conditions, the existence of more
undetected cloudy records in MODIS nighttime LST largely degrades the LST accuracy and
results in obviously lower $T_{min}$ estimation accuracy levels. However, why the $T_{min}$ estimations



clearly outperform $T_{max}$ under clear conditions (non-cloudy day condition) when both are free of
cloud effects remains unknown. One may argue that the so-called "clear" conditions are based on
only four satellite instantaneous observations and that actual cloudiness conditions may still be
cloudy. Although this is true, our study shows that even under clear conditions, the accuracy of
$T_{max}$ estimations based on daytime MODIS LST is much lower than those based on observed LST,
whereas the $T_{min}$ estimation based on nighttime MODIS LST shows comparable or even superior
accuracy.
From our previous analysis, we can attribute this difference in estimation accuracy between $T_{min}$
and $T_{max}$ to differences between daytime and nighttime MODIS LST. Much lower levels of
MODIS daytime LST accuracy than those for nighttime have been found in previous studies
(Krishnan et al., 2015;Min et al., 2015;Yu and Ma, 2011), and the validation tests shown in Sect.
4.2 also supports this conclusions. This precision bias is most likely attributable scale issues (Wan
et al., 2002;Wan, 2008). Single point measurements are difficult to make representative of the
1-km MODIS pixel when ground surfaces are complex (Coll et al., 2009;Hall et al., 2008). Many
studies have shown that MODIS daytime LST presents obviously lower levels of validation
accuracy than nighttime LST due to high levels of daytime LST heterogeneity (Wang et al.,
2008;Coll et al., 2009). In the daytime, cloud and hill shadows within pixels can produce
considerable LST heterogeneities while at night, the ground surface becomes cool and more
homogeneous when free of solar heating uncertainties (Wang et al., 2008). Oyler et al. (2016) also
show that daytime LST exhibits more spatial variation than $T_{air}$ while nighttime LST follows
similar spatial patterns as $T_{air}$ as demonstrated in his study.
In addition, it should be noted that clouds also have substantial effects on $T_{max}$ estimation. Thus, it
can be concluded that the frequently reported lower estimation accuracies of $T_{max}$ based on
MODIS daytime LST compared to those of $T_{min}$ based on nighttime LST (Zhu et al., 2013;Benali
et al., 2012;Zhang et al., 2011;Oyler et al., 2016) are mainly due to the mixed effects of \the
relatively low daytime LST accuracies and clouds.
To further prove this, four CMA stations (Fig. 9) presenting the largest reduction in RMSE values
after imposing clear conditions are selected for our $T_{min}$ and $T_{max}$ estimations. They can represent
practical application conditions where only daily meteorological observations can be obtained.
For $T_{max}$ estimation (Fig. 10), it is evident that forcing clear conditions has somewhat limited




effects on estimation performance. The samples collected under "cloudy day" conditions include
outliers far from the fit line derived using samples under "non-cloudy day" conditions. However,
the "non-cloudy day" samples still appear rather dispersed with many samples positioned far from
the fit line, and especially in the case of stations 89 and 41. This may illustrate   mixed effects of
both clouds and LST accuracies to some degree.
In contrast, the results of the $T_{min}$ estimation are somewhat inspiring. As shown in Fig. 11, a
number of cold-biased outliers that may be undetected cloudy records are captured by employing
cloudy conditions. More importantly, the "non-cloudy day" condition samples achieve a much
better fit. This not only demonstrates that undetected cloudy records are ubiquitous in MODIS
nighttime LST and that amounts can often be quite large but also that the influence of clouds on
$T_{min}$ estimations with true LST (i.e., without undetected clouds) is not substantial. Though the
actual cloudiness conditions are rather unpredictable and quite a few "good" samples around the
"non-cloudy day" fit line are also included in the "cloudy day" group, we consider constraining all
four MODIS observations for each day as non-cloudy as an efficient way to build a good fit
among $T_{min}$ estimations using MODIS nighttime LST as long as the amount of valid samples is
sufficient. This method can benefit studies requiring accurate $T_{min}$ estimations based on remotely
sensed LST.

**5.2 Uncertainty and error sources**
Emissivity issues may have caused the observed LST computation errors. Constant emissivity
values for the Ngari and Qinghai stations are used in our study, although this may not be
reasonable for non-growing seasons. However, the sensitivity experiments show that the influence
of emissivity values is not significant.
The $\leq 15$ min discrepancy may introduce uncertainties in data that intersect $T_{air}$, MODIS and
observed LST. Its influence is considered to be insignificant. Nighttime LST changes gently and
half-hourly observations can be used for MODIS LST validation as indicated in Wang et al.
(2008). $T_{air}$ also respond relatively slowly to LST, and MODIS daytime LST shows a strong
relationship to $T_{air}$ at a similar time discrepancy level ($\leq 12$ min) to that shown by Williamson et al.
(2013). Spatial heterogeneities within MODIS pixels of AWS may pose problems. As shown in
Fig. 1, such problems may not be severe, as land cover within the pixels of the three AWSs





appears to be largely homogeneous. The data quality of MODIS LST does not receive sufficient
consideration in this study. MODIS LST production involves the use of internal data quality flags,
and previous studies demonstrate that data quality is related to cloud contamination (Williamson et
al., 2013;Østby et al., 2014).
The validation accuracy of MODIS LST is affected by data quality (Krishnan et al., 2015).
However, rigid data quality constraints may severely decrease sample sizes due to relatively short
observation periods (1−2 years) used. This study presents results of general quality status, and
extreme low quality data (QC = 3) have been removed. Other factors including wind speeds and
sensor view zenith angles may affect results related to MODIS LST validation and the relationship
between $T_{air}$ and LST. According to Wang et al. (2008), the validation results are not or are weakly
affected by wind speed and the sensor view zenith angle. Wind speed has a limited effect on the
$T_{air}$-LST relationship, as shown by Gallo et al. (2011).
In addition, the results shown here are highly consistent across the three AWSs dominated by three
types of land cover, thus indicating that our results may be highly representative and that other
factors may not have played a key role in this study.

**6 Conclusion**
Cloud effects on $T_{min}$ and $T_{max}$ estimations according to MODIS LST are analyzed based on
detailed ground based observations drawn from three valuable AWSs and based on data drawn
from 92 CMA stations over the TP. Cloudiness is quantified using an efficient method based on
ground measurements of air temperature and downwelling longwave radiation. Comparisons made
between in-situ cloudiness observations and MODIS claimed clear-sky records shows that
erroneous rates of MODIS nighttime cloud detection are obviously larger than those for the
daytime. Our MODIS LST validation for different cloudiness constraining conditions reveals that
the accuracy of MODIS nighttime LST is severely affected by undetected clouds. However, the
accuracies of MODIS daytime LST do not seem to be influenced considerably by undetected
clouds.
Cloud effect tests show that $T_{min}$ estimations based on MODIS LST are mainly affected by large
errors introduced by undetected clouds in nighttime LST. However, clouds mainly influence $T_{max}$
estimation by affecting the relationship between $T_{max}$ and daytime LST. The effects of undetected



clouds in daytime LST are relatively weak. Frequently reported larger errors in $T_{max}$ estimations
based on daytime LST than those of $T_{min}$ based on nighttime LST may be largely attributed to
relatively large errors of MODIS daytime LST resulting from scale issues. Tests based on CMA
station observations further validate our results and show that constraining all four MODIS
observations per day as non-cloudy helps rule out undetected cloudy records while building good
$T_{min}$ estimation fit.
This study presents useful findings on the key effects of clouds on $T_{air}$ estimation based on
MODIS LST that can alleviate problems of severe data sparseness over the TP. More efficient
cloud detection methods for MODIS nighttime LST are needed for $T_{min}$ estimations. $T_{max}$
estimation based on daytime LST is rather challenging due to the complex effects of daily
cloudiness conditions in combination with scale issues.

**Author Contribution**
Professor Tian, He and Tang observed and provided the data of stations Nagri, Xiao Dongkemadi
and Qinghai, respectively. Professor Fan Zhang and Associate Professor Guoqing Zhang gave
many valuable suggestions to improve the manuscript. Hongbo designed the experiments and
wrote the manuscript.

**Acknowledgment**
This work was supported by the Chinese Academy of Sciences "Strategic Priority Research
Program (B)" (Grant No. XDB03030300); and by the National Natural Science Foundation of
China (Grant No. 41422101, 41271079, 41130638). We thank the Tanggula Station for Cryosphere
Environment Observation and Research and the Ngari Station for Desert Environment
Observation and Research for providing ground measurements of longwave radiation and air
temperature data. The Qinghai station data were downloaded from AsiaFlux (www.asiaflux.net).
We would like to thank Dr. Yanhong Tang for providing the ground measurements for the Qinghai
station. We are grateful to the Chinese Meteorology Administration for providing air temperature
data.




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



Table 1. Summary of the AWS sites

| AWS | Lon/Lat | Mean annual Precipitation (mm) | Mean annual air temperature (°C) | Elevation (m) | Land cover | Time period |
|---|---|---|---|---|---|---|
| Xiao Dongkemadi | 92.08/33.07 | 680 | -8.6 | 5621 | Glacier | 2009.1 - 2009.12 |
| Ngari | 79.70/33.39 | 125 | 1.2 | 4270 | Desert grassland | 2012.6 - 2013.12 |
| Qinghai | 101.30/37.60 | 567 | -1.7 | 3250 | Alpine meadow | 2003.1 - 2004.12 |






Table 2. Undetected MODIS LST clouds at 3 AWSs

| Site | Ratio of undetected cloudy records | | | |
|---|---|---|---|---|
| | Terra day (%) | Terra night (%) | Aqua day (%) | Aqua night (%) |
| Ngari | 5 | 3 | 3 | 15 |
| Xiao Dongkemadi | 12 | 15 | 11 | 37 |
| Qinghai | 3 | 20 | 3 | 50 |
| Average | 7 | 13 | 6 | 34 |





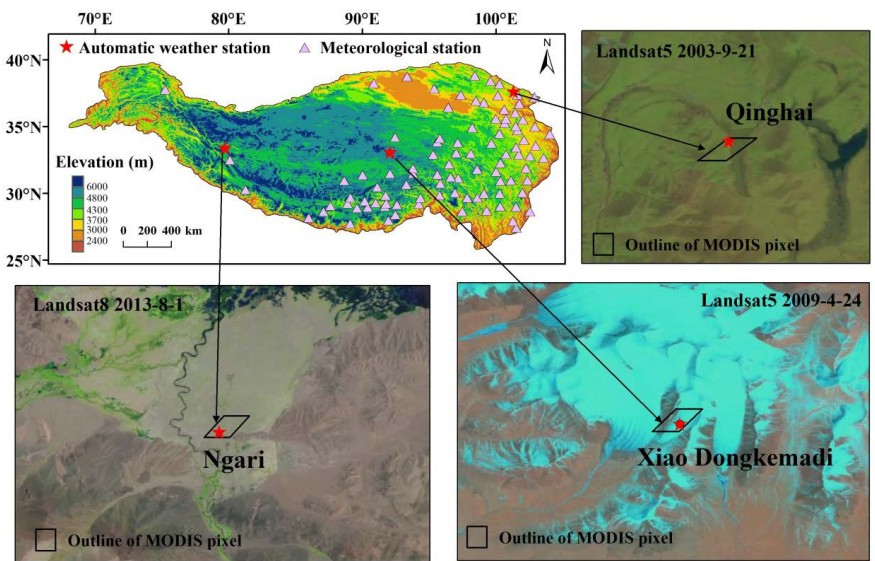



Figure 1: Map of the TP marking AWS and meteorological station locations. Landsat images
observed during the time period for data used in this study are also shwon in natural color modes
with capturing dates. The outline of the MODIS grid is also plotted.




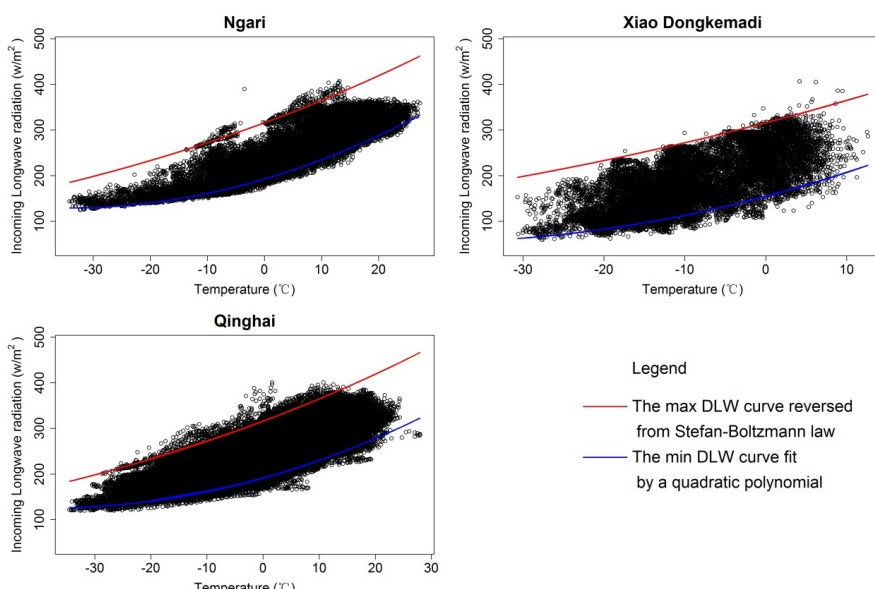


Figure 2: The distribution of downward longwave radiation (DLW) under different air

temperatures. The red line represents the max DLW curve reversed from the Stefan-Boltzmann

law. The blue line is the min DLW curve fitted by a quadratic polynomial.






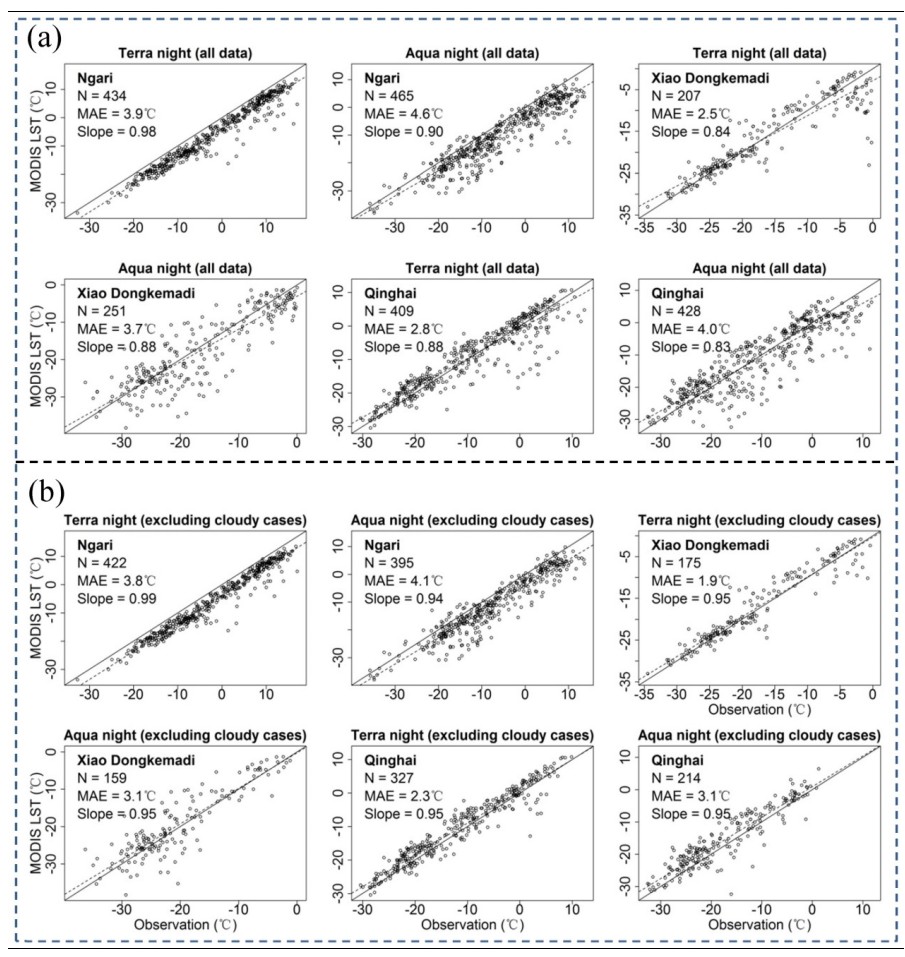



Figure 3: Validation of MODIS nighttime LST before (a) and after (b), excluding cloudy
cases.




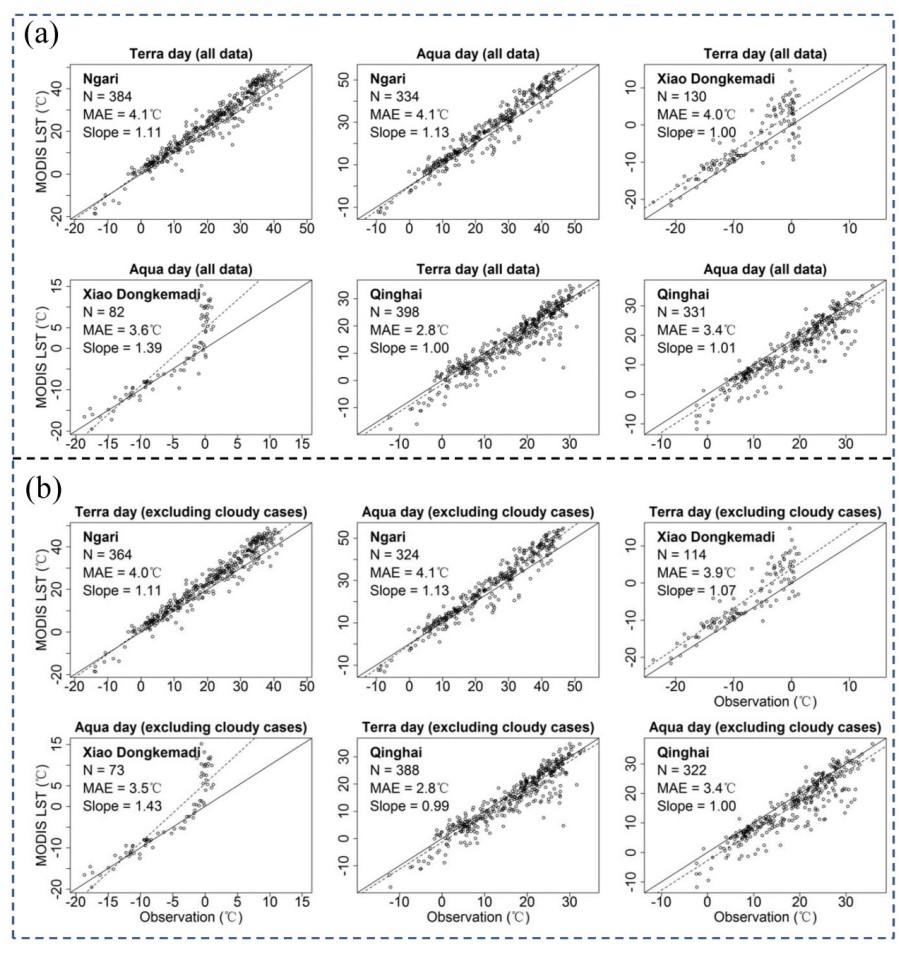


Figure 4: Validation of MODIS daytime LST before (a) and after (b), excluding cloudy cases.






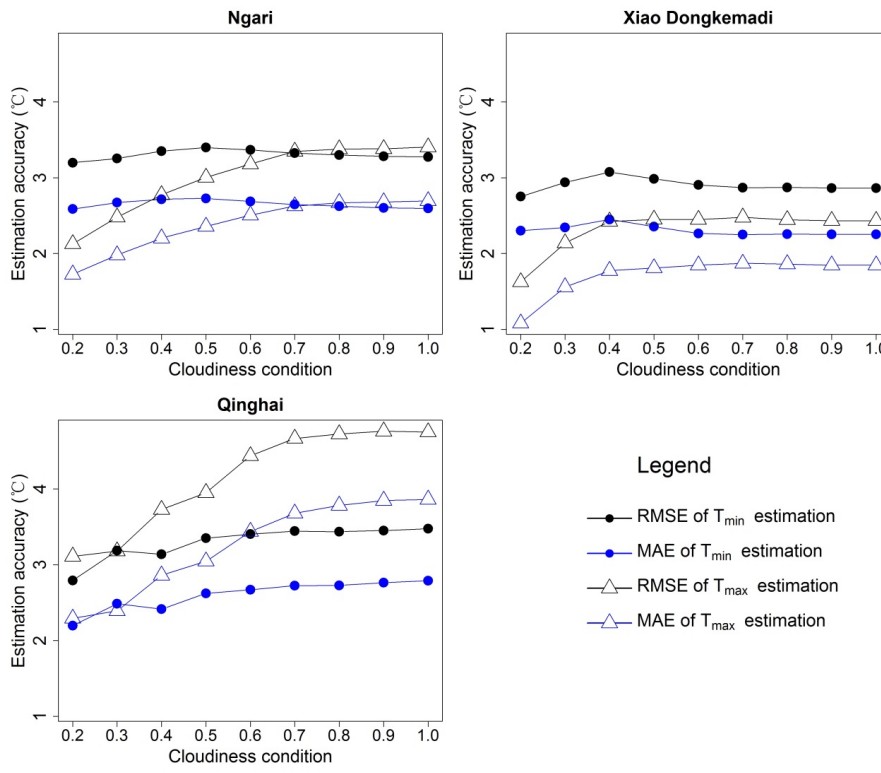



Figure 5: Accuracies (RMSE and MAE) of $T_{max}$ and $T_{min}$ estimations based on ground measured
LST under different cloudiness conditions across the three sites. The "cloudiness condition" is the
constraining condition of the daily averaged cloudiness index (CI). For example, a cloudiness
condition of 0.2 denotes a constraining daily mean of $CI \leq 0.2$.




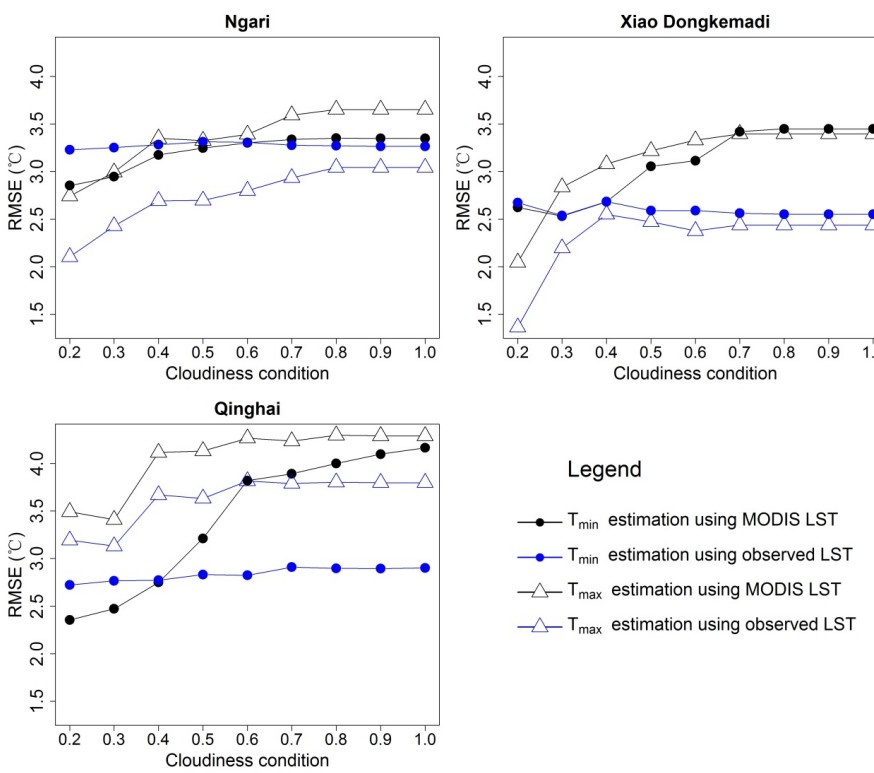



Figure 6: Accuracies (RMSE) of $T_{max}$ and $T_{min}$ estimations based on ground measured or MODIS

LST under different cloudiness conditions for the three AWSs. The "cloudiness condition" is the

constraining condition of the daily averaged cloudiness index (CI). For example, a cloudiness

condition of 0.2 denotes a constraining daily mean of $CI \leq 0.2$.






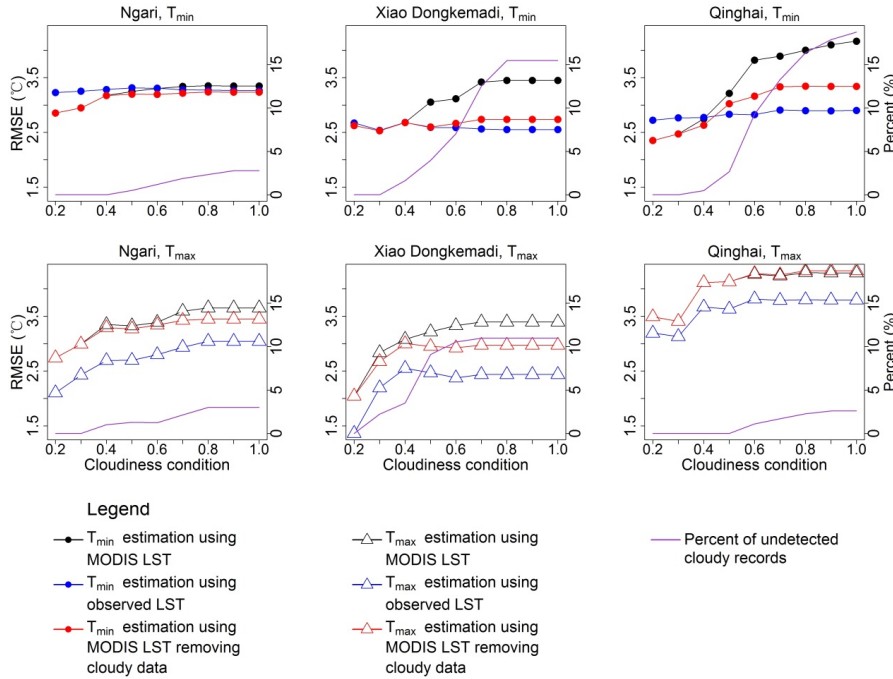


Figure 7: Comparisons between $T_{min}$ and $T_{max}$ estimation accuracies based on MODIS LST,
MODIS LST without cloudy data, and observed LST under different cloudiness conditions for the
three AWSs.





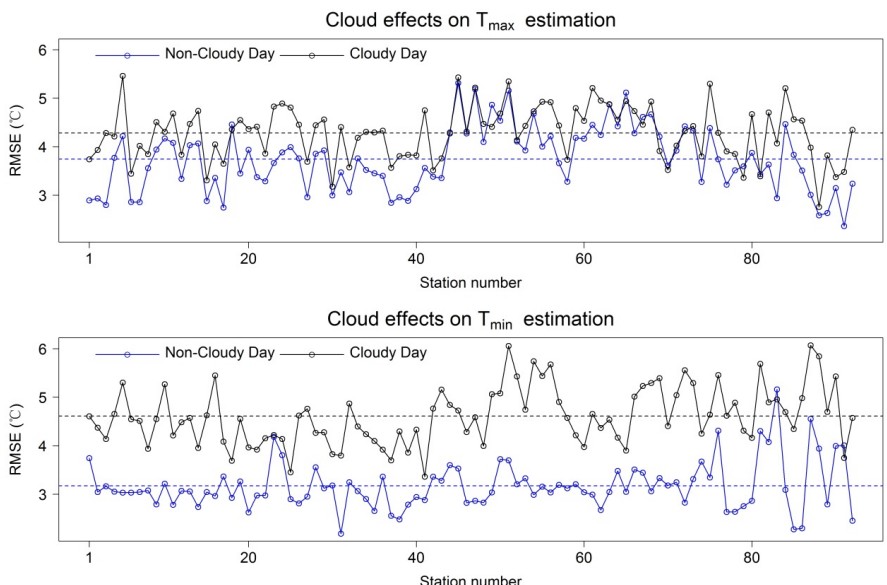



Figure 8: Comparisons of $T_{air}$ estimation accuracy levels based on MODIS LST and CMA
observations for "non-cloudy day" and "cloudy day" conditions.





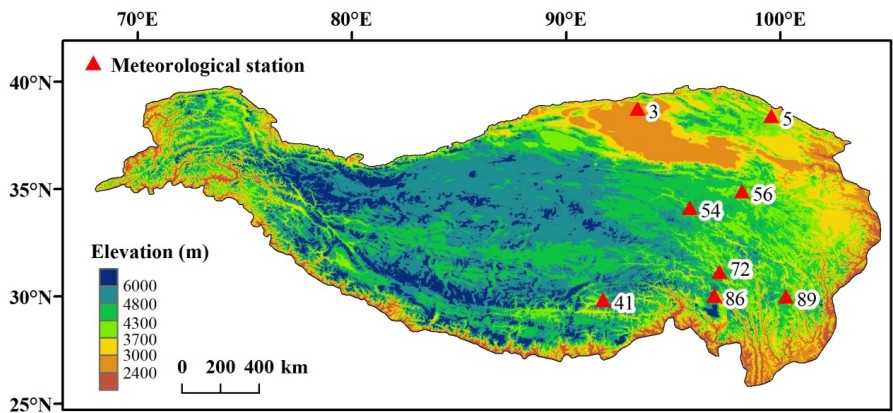

Figure 9: Locations of 4 representative CMA stations for $T_{min}$ (NO. 54, 56, 72, 86) and $T_{max}$ (NO.
3, 5, 41, 89) estimations.






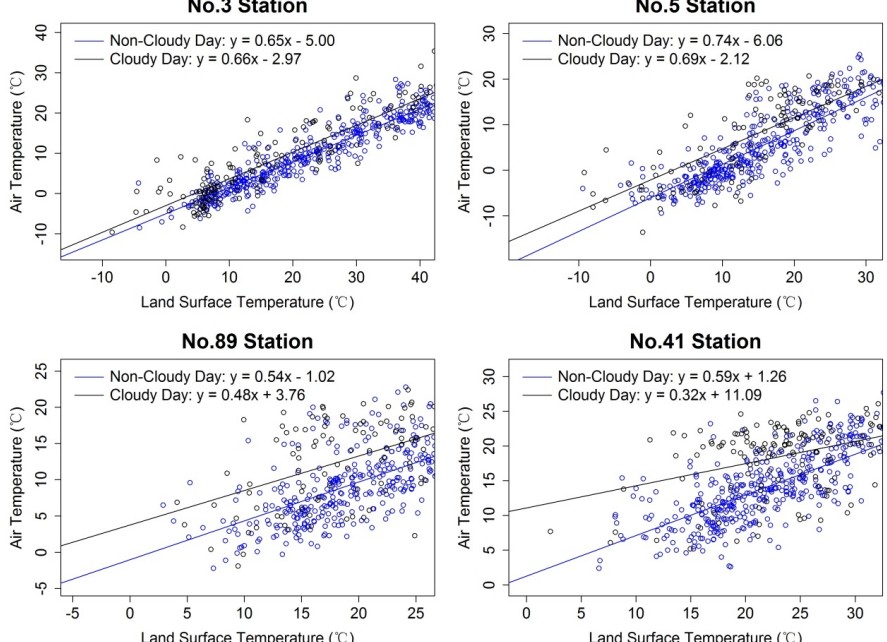



Figure 10: Comparisons of $T_{max}$ estimation accuracy between "cloudy day" and "non-cloudy day"
conditions at four meteorological stations presenting the largest decline in RMSE.





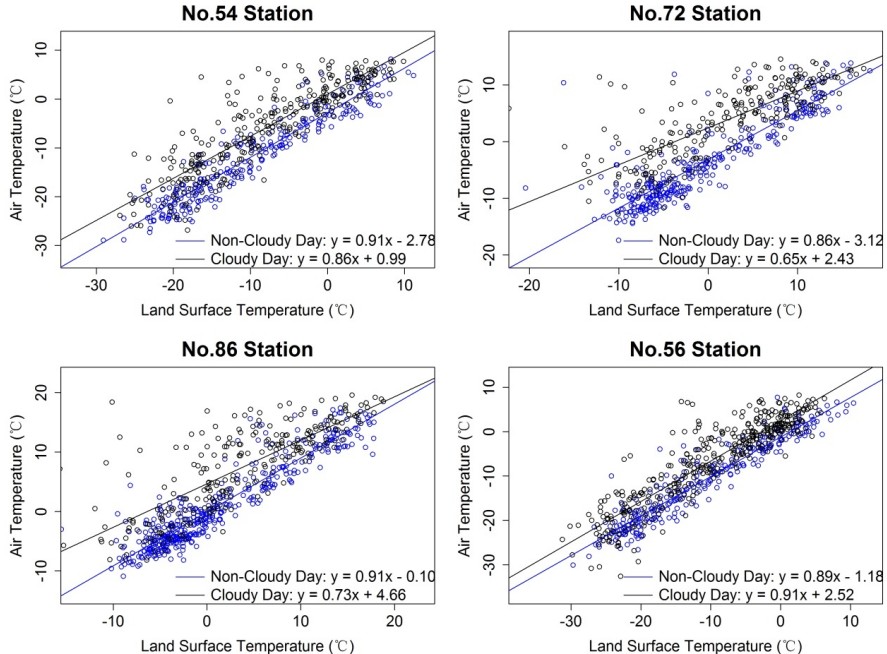



Figure 11: Comparisons of $T_{min}$ estimation accuracy between "cloudy day" and "non-cloudy day"
conditions at four meteorological stations presenting the largest decline in RMSE.
