# Peer review of "Published: 23 August 2016"

_Atmospheric Chemistry and Physics, 2016_

## Referee Comment (RC1) · Anonymous Referee #1 · 25 Sep 2016

This paper discusses the important effects of clouds on the relationship between air temperature and satellite LST. It gives a comprehensive analysis on how clouds affect the Tmax-Daytime LST and Tmin-Nighttime LST relations particularly for the LST data from MODIS, based on both AWS and CMA station data. The effects of undetected clouds on MODIS LST accuracies are first explored, and MODIS nighttime LST are found to receive much more negative effects than daytime. Then, the real Tmax-Daytime LST and Tmin-Nighttime LST relations are analyzed using observed LST, and clouds are found to have a much larger influence on Tmax estimation than Tmin. Further, MODIS LST and observed LST are used as proxies for estimating Tair respectively, and the results are compared. The authors conclude that for Tmin estimation,

large errors introduced by undetected clouds are key factors, while for Tmax, clouds strongly affect the relationship between Tmax and daytime LST. This study also discusses the clearly larger errors of Tmax than Tmin estimations and the heterogeneity of daytime LST is considered to be the main factor.

I think the authors have generally done good job of explaining their research and on the whole I found the paper reasonably straightforward to read. This paper is certainly worth of publication as it presents new and very useful information to researchers interested in estimating air temperatures from satellite data. However, there are few minor revisions that are required, as detailed below:

The abstract can be more concise. Some sentences should be condensed.

The order of references cited in the context appears to be a little mess, e.g. Line 53-55, Line 107-108, Line 177 . . . and many other lines. The authors should check and correct all of them.

In section 3.1: The way that "Ld is assumed to linearly increase from clear to overcast 185 conditions at a given temperature" may need a reference.

For section 3.3 "Tair estimation": The discussion about selection of linear regression as estimating method should be intensified.

Figure 3 and Figure 4: sub-plots should be plotted with the same scale.

Figure 5: When x > 0.4, the variation of Tmax estimating accuracy is very flat, especially for Xiao Dongkemadi. I think this should be discussed, possibly due to the sample amounts?

---

## Referee Comment (RC2) · Anonymous Referee #2 · 25 Sep 2016

**Review of "Evaluation of cloud effects on air temperature estimation using MODIS LST based on ground measurements over the Tibetan Plateau" by Zhang et al.**

In this paper, the authors evaluated the cloud effect on air temperature derived from MODIS land surface temperature based on ground measurements over the Tibetan Plateau. In summary, the authors revealed an interesting result. However, some questions and points need to be further addressed by some revisions before it can be published by ACP

The following is my comments:

(1) Line 86: A reference was missed, such as (Yu et al?).

(2) Line 144: Did you test the accuracy of LST derived from radiative transfer theory?

(3) Please show that the scattered points in the Fig.2 are based on the observed downward long-wave radiation. In addition, it is necessary to the further indicate how did you derive the cloud index in the section 3.1.

A reference is needed in Line 185.

(4) Section 3.2: My concern about the section is that subvisible cloud can affect the accuracies of MODIS LST, However, some aerosol layers also have a little bit effect, such as, at spring (Huang J., T. Wang, W. Wang, Z. Li, and H. Yan, 2014: Climate effects of dust aerosols over East Asian arid and semiarid regions. *Journal of Geophysical Research: Atmospheres*, 119, 11398–11416, doi:10.1002/2014JD021796.). How did you consider this issue?

(5) Section3.3: In your method, only LST was used to estimate the air temperature. Did you do some comparison with other methods? My main concern is that larger uncertainty maybe also exists in your method, thus some error evaluations are needed.

(6) In the section 3, a detailed flow chart is recommended, and can be make the paper

more clear.

---

## Author Comment (AC1) · 20 Oct 2016

The comment was uploaded in the form of a supplement:
http://www.atmos-chem-phys-discuss.net/acp-2016-747/acp-2016-747-AC1-supplement.zip
* * *

---

## Author Comment (AC2) · 20 Oct 2016

The comment was uploaded in the form of a supplement:
http://www.atmos-chem-phys-discuss.net/acp-2016-747/acp-2016-747-AC2-supplement.zip

---

## Author Response (AR1)

*A detailed, point-by-point response to the review comments is given below. Each review comment is repeated in **Bold** followed a description of our modification of the manuscript.*

**Anonymous Referee #1**

**This paper discusses the important effects of clouds on the relationship between air temperature and satellite LST. It gives a comprehensive analysis on how clouds affect the Tmax-Daytime LST and Tmin-Nighttime LST relations particularly for the LST data from MODIS, based on both AWS and CMA station data. The effects of undetected clouds on MODIS LST accuracies are first explored, and MODIS nighttime LST are found to receive much more negative effects than daytime. Then, the real Tmax-Daytime LST and Tmin-Nighttime LST relations are analyzed using observed LST, and clouds are found to have a much larger influence on Tmax estimation than Tmin. Further, MODIS LST and observed LST are used as proxies for estimating Tair respectively, and the results are compared. The authors conclude that for Tmin estimation, large errors introduced by undetected clouds are key factors, while for Tmax, clouds strongly affect the relationship between Tmax and daytime LST. This study also discusses the clearly larger errors of Tmax than Tmin estimations and the heterogeneity of daytime LST is considered to be the main factor.**

**I think the authors have generally done good job of explaining their research and on the whole I found the paper reasonably straightforward to read. This paper is certainly worth of publication as it presents new and very useful information to researchers interested in estimating air temperatures from satellite data. However, there are few minor revisions that are required, as detailed below:**

We greatly appreciate the reviewer's positive evaluation of our study. We have addressed all the detailed comments in the following.

**The abstract can be more concise. Some sentences should be condensed.**

Following this comment, some redundant statements in Abstract are deleted or integrated to make it more concise.

**The order of references cited in the context appears to be a little mess, e.g. Line 53-55, Line 107-108, Line 177 : : : and many other lines. The authors should check and correct all of them.**

All references in the context have been sorted in the order of "Year + Author".

**In section 3.1: The way that "Ld is assumed to linearly increase from clear to overcast 185 conditions at a given temperature" may need a reference.**

Thanks, the related references of Giesen et al., 2008; Yang et al., 2011; and Østby et al., 2014 have been added.

**For section 3.3 "Tair estimation": The discussion about selection of linear regression as estimating method should be intensified.**

We thank the reviewer for this valuable comment. Following this comment, section 3.3 has been rewritten, as "Various statistical methods have been used for $T_{air}$ estimation using MODIS LST, including neural network (Jang et al., 2004), random forests (Xu et al., 2014), M5 model tree (Emamifar et al., 2013) and the simple linear regression (Zhang et al., 2011;Benali et al., 2012;Lin et al., 2012). Comparisons among the performances of six types of statistical models with different levels of complexity for $T_{air}$ estimation indicate that though there truly exist some cases where advanced statistical models clearly outperform the simple linear regression model, the absolute differences of accuracies produced by different models are generally not big, especially for cases using MODIS nighttime LST (Zhang et al., 2016). Compared with the complex models such as neural network and random forests which introduce uncertainties owing to their much larger number of parameters, the linear regression model has the advantage of being easy to interpret and is most commonly used in previous studies (Zhang et al., 2011;Benali et al., 2012;Lin et al., 2012). In addition, an individual linear fit is built for each AWS or CMA station to make the relationship between $T_{air}$ and LST as locally accurate as possible and thus, variables indicating spatial coordinates (longitudes and latitudes) and land cover (e.g. NDVI) are not used. Therefore, the linear regression model using LST as the independent variable is chosen as the $T_{air}$ estimating method in this study."

**Figure 3 and Figure 4: sub-plots should be plotted with the same scale.**

Figures 3 and 4 have been replotted accordingly.

**Figure 5: When x > 0.4, the variation of Tmax estimating accuracy is very flat, especially for Xiao Dongkemadi. I think this should be discussed, possibly due to the sample**

**amounts?**

Yes, a sentence is added in section 4.3, as "It should be noted that when the "cloudiness condition" exceeds 0.6 (x > 0.6), the sample number no longer varies and due to the limited number of samples, the variation of $T_{max}$ and $T_{min}$ estimating accuracy is rather flat."

**Anonymous Referee #2**

**In this paper, the authors evaluated the cloud effect on air temperature derived from MODIS land surface temperature based on ground measurements over the Tibetan Plateau. In summary, the authors revealed an interesting result. However, some questions and points need to be further addressed by some revisions before it can be published by ACP**

We appreciate the reviewer's pertinent evaluations on our study very much. We have addressed all the detailed comments in the following.

**The following is my comments:**
**(1) Line 86: A reference was missed, such as (Yu et al?).**

Thanks. This reference has been added.

**(2) Line 144: Did you test the accuracy of LST derived from radiative transfer theory?**

Thanks. To reduce ambiguity, a sentence in section 2.1 in the revision is modified as "The LSTs of the Qinghai and Ngari stations were derived based on the Stefan–Boltzmann law and the thermal radiative transfer theory". To be clearer, a sentence is added in this section as, "The calculated LSTs were taken as ground measurements of LST as Wang et al. (2008)."

**(3) Please show that the scattered points in the Fig.2 are based on the observed downward long-wave radiation. In addition, it is necessary to the further indicate how did you derive the cloud index in the section 3.1.**
**A reference is needed in Line 185.**

Based on the comment, the caption of Fig.2 has been modified to show that the data points are observed values, as "The distribution of observed downward longwave radiation (DLW) under different air temperatures".

To further indicate, some descriptions are added in section 3.1, as "For example, for an observed downwelling longwave radiation as $L_i$ at the temperature $T_i$, if the $L_{max}$ and $L_{min}$ are the maximum and minimum $L_d$ under that temperature ($T_i$) respectively, then the CI is determined as $(L_i - L_{min}) / (L_{max} - L_{min})$."

The reference of Østby et al., 2014 describing the method for estimating cloud index is added.

**(4) Section 3.2: My concern about the section is that subvisible cloud can affect the accuracies of MODIS LST, However, some aerosol layers also have a little bit effect, such as, at spring (Huang J., T. Wang, W. Wang, Z. Li, and H. Yan, 2014: Climate effects of dust aerosols over East Asian arid and semiarid regions. Journal of Geophysical Research: Atmospheres, 119, 11398–11416, doi:10.1002/2014JD021796.). How did you consider this issue?**

We thank the reviewer for this comment. The effects of aerosol layers should be discussed. Some sentences are added in this section as "It should be noted that the effects of undetected clouds may come from or be mixed with the effects of residual/thin clouds (Platnick et al., 2003), fogs (Østby et al., 2014) and some thick aerosol layers (Huang et al., 2014) existing in the MODIS pixel, which may impose errors on the MODIS LST product to varying degrees. Even though these effects are hard to distinguish in detail, undetected clouds are generally considered to have strong negative effects on the accuracies of MODIS LST (Williamson et al., 2013;Østby et al., 2014;Shamir and Georgakakos, 2014)."

**(5) Section3.3: In your method, only LST was used to estimate the air temperature. Did you do some comparison with other methods? My main concern is that larger uncertainty maybe also exists in your method, thus some error evaluations are needed.**

We thank the reviewer for this valuable comment. In fact, we compared the performances of six statistical methods for daily air temperature estimation in another work of us recently published (Zhang et al., *in press*). Following this comment, section 3.3 has been rewritten, as "Various statistical methods have been used for $T_{air}$ estimation using MODIS LST, including neural network (Jang et al., 2004), random forests (Xu et al., 2014), M5 model tree (Emamifar et al., 2013) and the simple linear regression (Zhang et al., 2011; Benali et al., 2012; Lin et al., 2012). Comparisons among the performances of six types of statistical models with different levels of complexity for $T_{air}$ estimation indicate that though there truly exist some cases where advanced statistical models clearly outperform the simple linear regression model, the absolute differences of accuracies produced by different models are generally not big, especially for cases using MODIS nighttime LST (Zhang et al., *in press*). Compared with the complex models such as neural network and random forests which introduce uncertainties owing to their much larger number of parameters, the linear regression model has the advantage of being easy to interpret and is most commonly used in previous studies (Zhang et al., 2011; Benali et al., 2012; Lin et al., 2012). In addition, an individual linear fit is built for each AWS or CMA station to make the relationship between $T_{air}$ and LST as locally accurate as possible and thus, variables indicating spatial coordinates (longitudes and latitudes) and land cover (e.g. NDVI) are not used. Therefore, the linear regression model using LST as the single independent variable is chosen as the $T_{air}$ estimating method in this study."

**(6) In the section 3, a detailed flow chart is recommended, and can be make the paper more clear.**

We thank the reviewer for this valuable comment. A flow chart is added as below:

[revised manuscript text omitted]